# Empowering high-dimensional optical fiber communications with integrated photonic processors

Kaihang Lu [1,4], Zengqi Chen [1,4], Hao Chen [1,4], Wu Zhou [1], Zunyue Zhang [2,3], Hon Ki Tsang [2] ✉ & Yeyu Tong [1] ✉

Mode-division multiplexing (MDM) in optical fibers enables multichannel capabilities for various applications, including data transmission, quantum networks, imaging, and sensing. However, high-dimensional optical fiber systems, usually necessity bulk-optics approaches for launching different orthogonal fiber modes into the optical fiber, and multiple-input multiple-output digital electronic signal processing at the receiver to undo the arbitrary mode scrambling introduced by coupling and transmission in a multi-mode fiber. Here we show that a high-dimensional optical fiber communication system can be implemented by a reconfigurable integrated photonic processor, featuring kernels of multichannel mode multiplexing transmitter and all-optical descrambling receiver. Effective mode management can be achieved through the configuration of the integrated optical mesh. Inter-chip MDM optical communications involving six spatial- and polarization modes was realized, despite the presence of unknown mode mixing and polarization rotation in the circular-core optical fiber. The proposed photonic integration approach holds promising prospects for future space-division multiplexing applications.

The spatial dimension of optical fibers remains an untapped resource for enhancing their information-transmission capacity[1–4]. Space-division multiplexing (SDM), whereby multiple data signals are multiplexed into different spatial channels, has attracted much research interest, including the use of multiple single-mode cores sharing a common cladding or multiple orthogonal modes in a multi-mode optical fiber. SDM fiber can thus be classified into multi-core fiber (MCF), few-mode fiber (FMF), and multi-mode fiber (MMF)[1,5,6]. However, leveraging the spatial dimension of optical fibers can be very challenging, particularly when higher-order modes are involved in an FMF or MMF for mode-division multiplexing (MDM) systems. Two major challenges are associated with MDM optical fiber systems, including the lack of low-cost and scalable mode (de)multiplexers that can generate or decouple multiple orthogonal fiber modes and the substantial energy

consumption and large time latency needed to descramble mixed optical signals through electronic digital signal processing (DSP)[7].

Previous approaches for implementing mode (de)multiplexers include the use of optical phase plates[8,9], spatial light modulators (SLMs)[10], or multi-plane light conversion (MPLC)[11–14]. Although substantial advancements have been achieved in fiber-based photonic lanterns or laser-inscribed waveguides[5,6,15], a compact and low-cost approach is desired, especially for short-reach optical communications within data centers where cost and footprint are crucial factors. Moreover, even in the absence of any disturbances, when light travels through a circular-core MMF, various speckle patterns can be formed due to differing dephasing conditions between the fiber eigenmodes[16–19]. Consequently, the one-by-one mapping of fiber modes between the transmitter side and the receiver side becomes

[1]Microelectronic Thrust, The Hong Kong University of Science and Technology (Guangzhou), 511453 Guangzhou, Guangdong, PR China. [2]Department of Electronic Engineering, The Chinese University of Hong Kong, Shatin, New Territories, 999077 Hong Kong, PR China. [3]School of Precision Instrument and Opto-Electronics Engineering, Tianjin University, 300072 Tianjin, PR China. [4]These authors contributed equally: Kaihang Lu, Zengqi Chen, Hao Chen.
✉e-mail: hktsang@ee.cuhk.edu.hk; yeyutong@hkust-gz.edu.cn

challenging. Complex speckle patterns at the fiber end can cause significant and unknown scrambling of the information encoded on different orthogonal fiber modes. Rectangular-core FMFs have thus been proposed to break rotational symmetry and prevent spatial degeneracy[18,19]. However, inter-mode coupling can also occur due to factors such as fiber non-uniformity, sharp bending, mechanical misalignment of fiber splices, or imperfect mode (de)multiplexers. Essentially, the information is not lost, but separating the arbitrarily mixed signals in a high-dimensional fiber system presents a significant challenge. Previous studies have demonstrated that this issue can be effectively addressed through coherent communications and digital electronic multiple-input multiple-output (MIMO) processing[8,20]. However, this method, originally designed for wireless communications, requires extremely high-speed digital circuits for optical fiber communications. The resulting high power consumption and large time latency at high data rates become concerns. Moreover, increasing the number of spatial channels would also rapidly increase the dimensionality of the MIMO equalizer, thereby further increasing DSP complexity and impeding its potential utilization[21].

By transforming the 2D field profile of the optical fiber into the planar waveguide modes on the integrated photonic chip, such as through grating couplers[17,22–26], photonic processors present a promising alternative technology for managing the high-dimensional optical fiber system, especially on the silicon photonics platform, which offers low-cost, high-volume manufacturing with CMOS compatibility[27–29]. The high refractive index contrast of the silicon photonics platform enables ultra-compact confinement of optical field for high-density and multichannel optical input/output (I/O)[17,23–26]. Meanwhile, integrated coherent optical mesh has been demonstrated for implementing arbitrary matrix transformations in optical neural networks[30–32], reconfigurable signal processors[33–36], free-space and on-chip beam separation[37–39], and quantum networks[40,41]. By integrating multimode optical I/O and optical matrix processing on the same chip, photonic processors have the potential to offer an enabling technology for MDM optical fiber systems.

In this work, we developed a reconfigurable integrated photonic processor capable of selectively launching and separating orthogonal optical fiber modes. High-dimensional chip-to-chip optical fiber communications can be directly achieved with a two-mode FMF involving the full set of six LP modes, including $LP_{01-x}$, $LP_{01-y}$, $LP_{11a-x}$, $LP_{11a-y}$, $LP_{11b-x}$, and $LP_{11b-y}$. Selective mode excitation in the optical fiber is performed at the transmitter side using an efficient and multimode optical I/O. To address the unknown mode scrambling and polarization rotation after fiber transmission at the receiver end, a reconfigurable Mach Zehnder interferometer (MZI) based optical mesh is employed to apply inverse matrix transformations of the optical fiber and function as an all-optical MIMO descrambler. A high-dimensional optical fiber communication system managed by the integrated silicon photonic processor is experimentally demonstrated.

## Results

### Integrated photonic processor design

Figure 1a illustrates the high-dimensional optical fiber communication system enabled by the silicon photonic integrated circuits (PICs). The optical modes in a circular-core FMF can be described by eigenmodes with a rigorous vectorial treatment of the wave equation in cylindrical coordinates. LP mode, shown in Fig. 1b, is a description often used for the linearly polarized superposition of fiber eigenmodes. At the transmitter side, the chip-to-fiber coupling is realized by an efficient and multimode 2D grating coupler, as depicted by Fig. 1c. By controlling the relative phase delay between the two counter-propagating quasi-transverse-electric (TE) modes on chip, all the six spatial and polarization channels in a two-mode FMF can be selectively launched, including $LP_{01-x}$, $LP_{01-y}$, $LP_{11a-x}$, $LP_{11a-y}$, $LP_{11b-x}$, and $LP_{11b-y}$. The design of the multimode grating coupler and the selective mode launching are explained in Supplementary Note 1. It is worth noting that although selective decoupling is desired at the receiver side, this can only happen when the fiber LP modes are of high modal purity and polarizations are accurately aligned. In reality, LP mode deformation and polarization rotation are inevitable in a circular-core FMF, which results in an unpredictable field pattern arriving at the coupling end between the fiber and the photonic chip[16–18]. For example, upon launching the $LP_{11a}$ mode into the FMF from the transmitter side, the resulting speckle pattern typically manifests as a linear combination of all $LP_{11}$ spatial and polarization modes, which

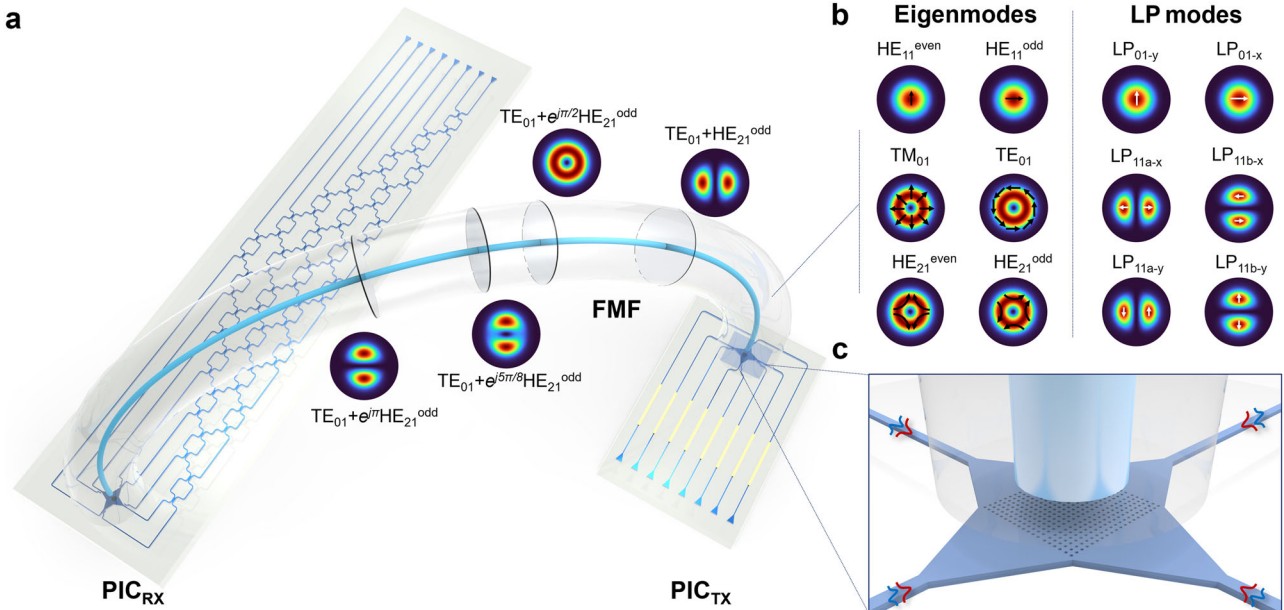

**Fig. 1 | System schematic diagram. a** High-dimensional optical fiber communication system with reconfigurable integrated photonic processor. $PIC_{TX}$: photonic integrated circuits at the transmitter side; $PIC_{RX}$: photonic integrated circuits at the receiver side. Optical signals in different modes may experience mixing as they propagate through a circular-core few-mode fiber (FMF), owing to rotation symmetry and spatial degeneracy. **b** Mode field profile of the eigenmodes and the linear polarized (LP) modes in a two-mode FMF. **c** Integrated multimode grating coupler serving as the optical I/O for FMF (not to scale).

undergo continuous mixing along propagation in the optical fiber, as illustrated in Fig. 1a. This is due to the fact that LP modes are essentially formed by linear combinations of the fiber eigenmodes with varying interference conditions. Nevertheless, the proposed 2D multimode grating coupler can support all the six eigenmodes in a two-mode FMF, which allows non-selective decoupling of the multimode optical signals into the eight single-mode channels on chip. The transmission matrix of the multimode grating coupler at the receiver side can thus be denoted by an 8 by 6 matrix $A_{8\times6}$. Efficient chip-to-fiber and fiber-to-chip coupling can be obtained with a small mode-dependent loss, as shown by the simulation results in Figure S2. In order to mitigate the unknown inter-modal signal mixing, a reconfigurable optical mesh with a dimensionality of eight can be programmed to apply the inverse transformation of the fiber transmission matrix, thereby recovering the six orthogonal channels permitted in the two-mode FMF. The transformation matrix $M$ from the transmitter to the receiver can be formulated as follows:

$$M = U_{8\times8}A_{8\times6}U_{6\times6}\mathrm{pinv}(A)_{6\times8} \tag{1}$$

where $U(8)$ and $U(6)$ denote the transformation matrix of the unitary optical mesh and the two-mode FMF, respectively. The transformation matrix of the multimode grating coupler at the transmitter side can be expressed as the pseudo-inverse of the receiver side, denoted as $\mathrm{pinv}(A)_{6\times8}$. A comprehensive explanation of the matrix operations of the photonic processor is included in Supplementary Note 2. In this demonstration, a unitary optical mesh $U(8)$ is utilized, taking advantage of the low channel-dependent loss of the multimode grating coupler. More complex optical meshes would be necessary to improve the system performance with severe loss difference or differential mode group delay[42-44].

Figure 2a shows the scanning electron microscope image of the 2D grating coupler with 70-nm shallowly etched circular holes. The experimental coupling loss spectra of $LP_{01}$, $LP_{11a}$, and $LP_{11b}$ in the two orthogonal polarizations for a two-mode FMF are measured and presented in Fig. 2b. The x-polarized $LP_{01}$, $LP_{11a}$, and $LP_{11b}$ exhibit a peak experimental efficiency of −3.5 dB, −6.1 dB, and −4.3 dB at 1532 nm, 1517 nm, and 1515 nm respectively. As the 2D grating utilizes a symmetric structure, similar coupling efficiencies for the y-polarized modes can be obtained, measuring −3.9 dB at 1527 nm for $LP_{01y}$, −3.9 dB at 1517 nm for $LP_{11a-y}$, and −6.1 dB at 1525 nm for $LP_{11b-y}$, respectively. To validate the selective launching of the six LP modes through the multimode grating coupler, the diffracted optical field of the grating is captured using a 10× microscope objective and an

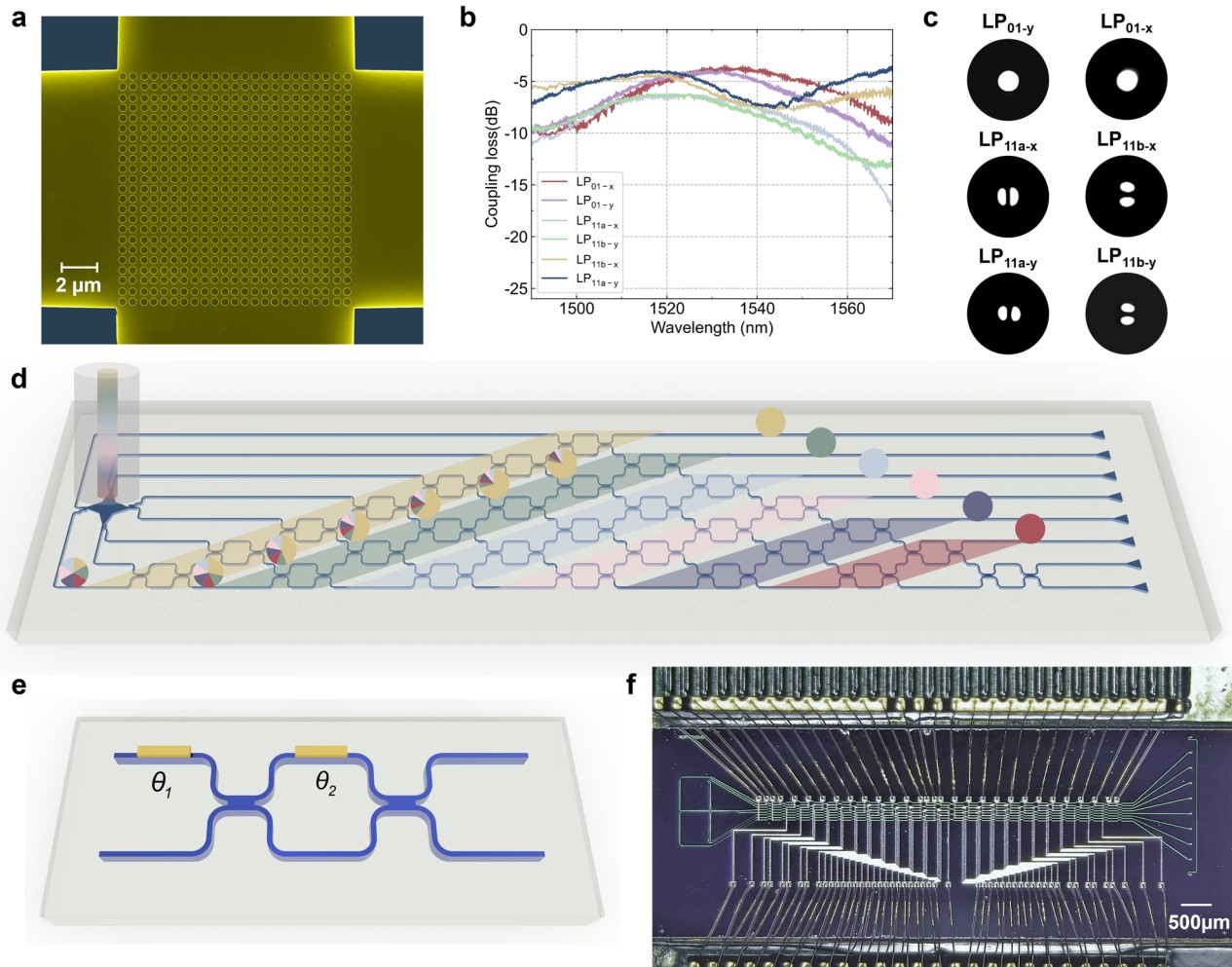

Fig. 2 | Multimode grating coupler characterization and photonic processor design. a Scanning electron microscope (SEM) image of the 2D grating coupler. b Experimental chip-to-fiber coupling efficiency spectra of the multimode grating coupler for various LP modes. c Optical field profile of the grating coupler captured by an infrared camera with a 10× microscope objective when different fiber mode is selectively launched by the $\mathrm{PIC_{TX}}$. d Schematic of the optical mesh based on Mach-Zehnder interferometers (MZIs) at the receiver side for mode unscrambling in the optical domain. e 2 × 2 unitary operation based on the MZI with two optical phase shifters. $\theta_1$ and $\theta_2$ denote the phase shift induced by the outer and inner thermal phase shifter, respectively. f Microscopic image of the wired-bonded integrated photonic processor used at the received side.

infrared camera, as depicted in Fig. 2c. Table S1 summarizes a comparison of the reported multimode optical I/O for two-mode circular-core FMFs. The diagram of the photonic processor at the receiver side is shown in Fig. 2d. It includes a multimode grating coupler, tapered asymmetric directional couplers (ADCs), MZI-based linear unitary matrix, and eight output single-mode grating couplers. The $8 \times 8$ triangular optical mesh is formed by 28 tunable MZIs, as depicted in Fig. 2e. Figure 2f shows the microscopic image of the wire-bonded photonic processor at the receiver end. The total footprint of the photonic integrated circuits is 8.5 mm × 1.8 mm.

## Mode descrambling experiment

With various fiber modes selectively launched, the FMF is directly bridged with the photonic chip at the receiver side. Figure 3a shows the experimental setup for mode descrambling with a 5-meter FMF using the integrated photonic processor. Because of the mode deformation and polarization rotation, the received speckle pattern at the fiber-to-chip end is uncertain, which results in a random and unknown received optical power distribution entering the optical mesh. A fiber array unit (FAU) and multichannel power meter are used to monitor the optical power from the eight output ports of the optical mesh. The

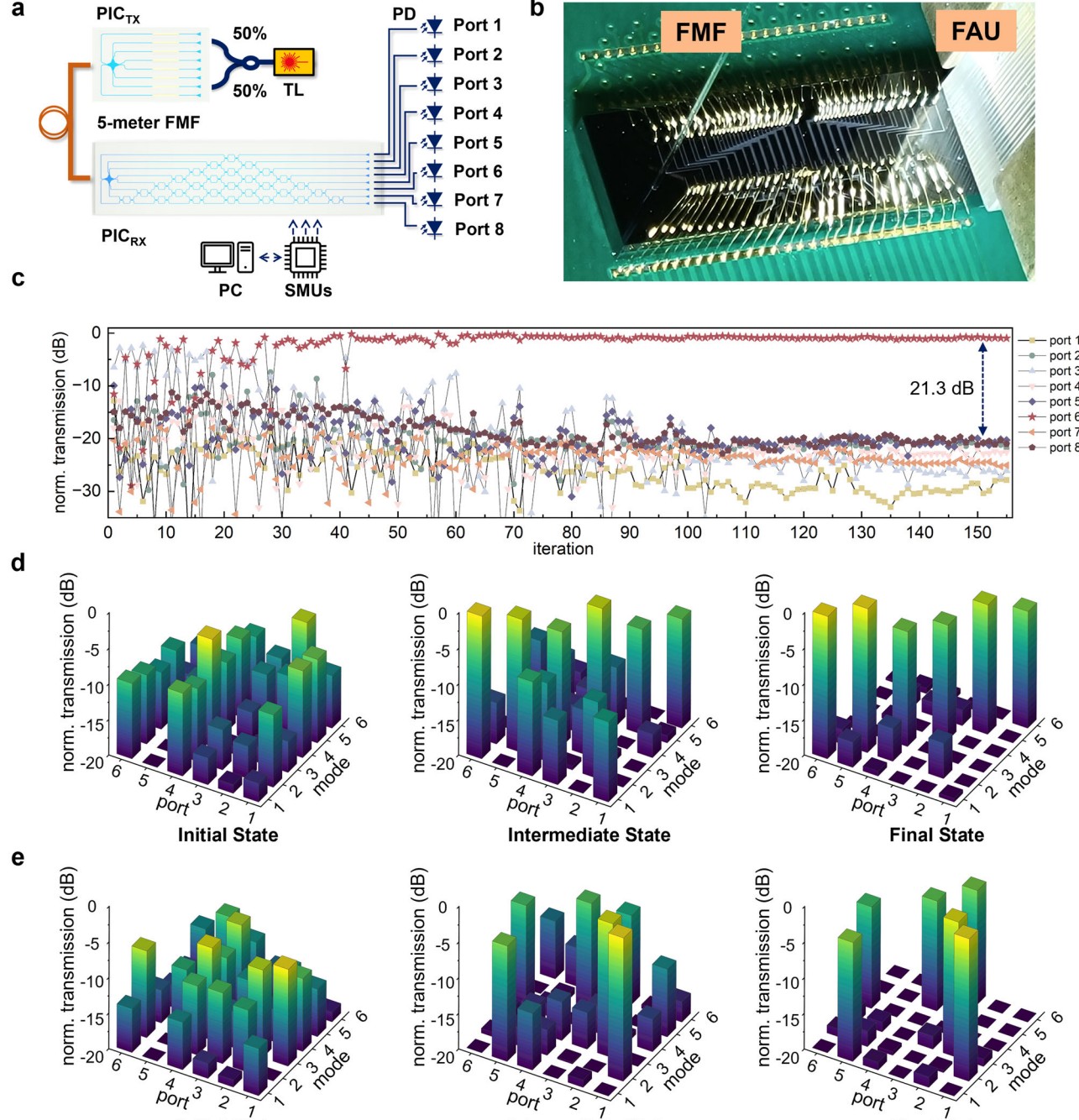

**Fig. 3 | Photonic processor configuration and mode descrambling results.**
**a** Experimental setup used for inter-chip optical mode selective launching and descrambling. TL: tunable laser, BS: beam splitter, FMF: few-mode fiber, PD: photodiode. **b** Photograph of the wired-bonded photonic chip under test at the receiver side with FAU (fiber array unit) and FMF. **c** Evolution of the normalized transmission for eight output ports during configuration of the optical mesh. **d** Bar chart of the initial random state, intermediate state, and final state for 6 spatial and polarization channels during optical mesh configuration. **e** Bar chart of another routing configuration of the optical mesh. The mode 1-6 refer to $LP_{01-x}$, $LP_{01-y}$, $LP_{11a-x}$, $LP_{11a-y}$, $LP_{11b-x}$, $LP_{11b-y}$.

photograph of the photonic chip mounted on a printed circuit board at the received side under test is presented in Fig. 3b. A thermoelectric cooler (TEC) is used for thermal stabilization of the integrated photonic processor. The detailed optical loss breakdown of the optical system is summarized in Supplementary Note 3. In the experiment, we have employed multichannel source measurement units (SMUs) and the particle-swarm optimization[45] algorithm to optimize the drive voltage of phase shifters within the optical mesh $U(8)$. The figure-of-merits has been defined as the minimum crosstalk suppression between the target output port and the other output ports. Once an acceptable solution is identified or the maximum number of iterations is reached, the optimization algorithm terminates and locks the unscrambling status by fine-tuning the driving voltage of each phase shifter inside the optical mesh.

Figure 3c presents the evolution of normalized optical power from eight output ports during configuration. It is evident that after around 120 iterations, selective decoupling can be configured with a maximum crosstalk level suppression of ≥21.3 dB. Through the same experimental setup, six channels, including the two orthogonal polarizations of the $LP_{01}$, $LP_{11a}$, and $LP_{11b}$ modes are selectively launched and decoupled with transmission matrices bar chart summarized in Fig. 3d. The initial power randomization for all fiber modes can be configured to different output ports, with experimental power isolation ratios all above 15.2 dB. Different routing schemes of fiber modes are also implemented to validate the configurability of our photonic processor, showing a similar performance as depicted by Fig. 3e. Temperature sensitivity of the integrated photonic processor is also evaluated. When the temperature variation is kept below 2.5 °C, the increase in inter-modal crosstalk can be limited to 3 dB. The optical mesh can be reconfigured to avoid crosstalk degradation, highlighting the need for real-time configuration in the future.

## MDM optical fiber communications

To evaluate the high-speed communication performance of the reconfigurable photonic processor, 32 Gbps non-return-to-zero (NRZ) signals are generated at the transmitter side by a bit pattern generator (BPG) and an external $LiNO_3$ Mach-Zehnder modulator. The pseudorandom binary sequence (PRBS) has a period length of $2^{15}-1$. Because the single-mode grating coupler array in this work is also centered at 1535 nm with a coupling loss of about −4.5 dB. The communication system is operated at 1530 nm to reduce the transmission loss and mode-dependent loss while within the working wavelength range of the erbium-doped fiber amplifier. The experimental setup using the integrated photonic processor to selectively launch and decouple various LP mode carrying optical signals is shown in Figure S4a. After configuring an integrated photonic processor, various orthogonal fiber modes can be launched and routed to any desired output port to undo the signal-mixing process. Figure S5 presents clear and open-eye diagrams for each of the fiber LP modes, which primarily benefited from the low-loss multichannel optical I/O. The experimental results reveal that a high-dimensional optical fiber communication system can be realized by the reconfigurable integrated photonic processor.

Additional concurrent data channels are also launched from the transmitter side to further assess the performance of all-optical descrambling. Because our input grating couplers at the transmitter side are not aligned with the pitch size of the FAU (as depicted in Fig. S4b), we use a three-port mode-selective fiber photonic lantern[46] to launch various LP modes simultaneously. Figure S4c depicts the experimental setup with two concurrent data channels injected using two orthogonal fiber modes, including the combination of $LP_{01}$ with $LP_{11a}$, $LP_{01}$ with $LP_{11b}$, and $LP_{11a}$ with $LP_{11b}$. The optical mesh at the receiver side is first configured for mode decoupling until the minimum of crosstalk between the two channels can be reached. The two-by-two bar charts of the normalized received optical power for each combination are illustrated in Fig. 4a–c. The optimal inter-modal

crosstalk obtained ranges between −24 dB to −29 dB. Figure 4d presents the retrieved eye diagram of a single channel as a reference. When two concurrent modes are launched without configuring the optical mesh, the eye diagram becomes completely closed, as shown in Fig. 4e. This is attributed to the coherent interference between the spectrally overlapped channels. By applying the corresponding configuration of the optical mesh shown in Fig. 4a, $LP_{01}$ and $LP_{11a}$ can be separated, with the resulting eye diagrams and bit error rates (BERs) shown in Fig. 4f. A small BER penalty is observed compared to a reference signal with no concurrent channels injected, with the penalty being proportional to the inter-modal crosstalk level. The corresponding eye diagrams and BERs for simultaneous launching of $LP_{01}$ with $LP_{11b}$ and $LP_{11a}$ with $LP_{11b}$ are shown in Figs. 4g and 4h, respectively.

As the number of concurrent channels increases, the accumulated inter-modal crosstalk would further degrade the BER performance. This can be validated by injecting three concurrent channels utilizing the experimental setup shown in Figure S5d. To obtain three spatially decoupled $LP_{01}$, $LP_{11a}$, and $LP_{11b}$ modes, 2-km and 5-km SMFs are used before launching the optical signals into the fiber photonic lantern. An optimal crosstalk suppression level >21 dB can be obtained. Figure 5a presents the closed eye diagram resulting from mode scrambling, along with the retrieved eye diagrams when only a single channel, two channels, or three channels are launched from the transmitter side. BER measurements are also performed with power penalty illustrated in Fig. 5b.

## Discussion

We demonstrated a high-dimensional optical fiber communication system enabled by a reconfigurable silicon photonic processor. This system incorporates selective mode excitation and all-optical mode descrambling, achieved physically through the use of multimode optical I/O and optical mesh based on silicon MZIs. Without prior knowledge of the random mode mixing and polarization rotation during the circular-core FMF transmission, we effectively reconstruct six spatial and polarization channels, including the full set of orthogonal channels in a two-mode FMF for chip-to-chip optical fiber communications. As the mode launching and unmixing are all conducted in the optical domain, our approach is expected to be generally capable of managing communication systems utilizing various modulation formats. In real MDM optical fiber system, mapping individual modes presents a challenge due to the unknown mode mixing resulting from factors such as mode deformation, bending, or even structural imperfections in the optical fiber. However, programmable photonic processors hold the potential to effectively manage all the spatial channels prior to photodetection, thereby opening the door to numerous high-dimensional optical fiber applications, including communications, quantum networks, imaging, and sensing.

Complete singular value decomposition (SVD) would be needed in the future to reduce the crosstalk level and support more concurrent orthogonal data channels with a small BER penalty. Although it would require a more complex optical mesh to execute non-unitary linear matrix transformations, unraveling optical modes affected by significant mode-dependent loss and differential mode group delay will be possible for multimode optical fiber systems[42–44].

Increasing the number of optical channels can be achieved by harnessing the available optical bandwidth of the integrated photonic processor. Each optical fiber mode can then be utilized in conjunction with wavelength-division multiplexing to enhance the overall data throughput. Apart from that, the number of involved spatial channels can also be increased by optimizing the multimode optical I/O and the optical mesh. For instance, $LP_{21}$ mode can also be launched by feeding two counterpropagating $TE_1$ modes with a relative phase shift of π, using the same multimode grating coupler at the transmitter side in this work[26]. To undo the signal mixing, however, a non-unitary optical

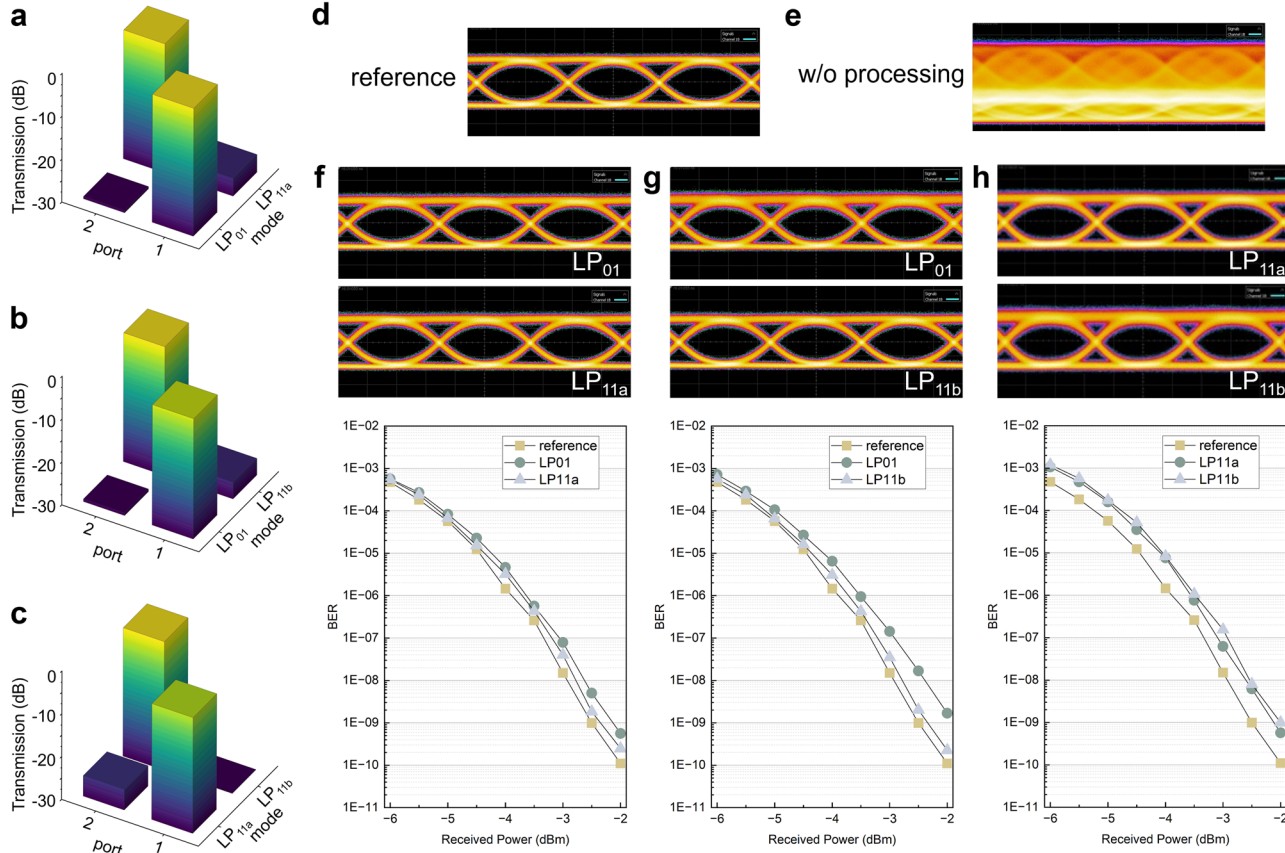

**Fig. 4 | Experimental results for mode-division multiplexing communications with two concurrent channels.** Bar chart of the normalized received optical power for **a** mode $LP_{01}$ and $LP_{11a}$, **b** mode $LP_{01}$ and $LP_{11b}$, **c** mode $LP_{11a}$ and $LP_{11b}$. **d** Reconstructed eye diagram without any concurrent channel as a reference. **e** Closed eye diagram due to coherent beating without photonic processing. Eye diagrams and bit error rates (BERs) after decoupling two concurrent mode channels, including **f** mode $LP_{01}$ with $LP_{11a}$ **g** mode $LP_{11a}$ with $LP_{11b}$, **h** mode $LP_{11a}$ with $LP_{11b}$.

mesh would be required as not all of the degenerate modes in the $LP_{21}$ group can be efficiently coupled back to the photonic chip. This may also require increasing the dimension size of the integrated optical mesh or positioning it at the transmitter side[43,47].

Real-time configuration of the integrated photonic processor is crucial for high-dimensional fiber systems, as the transmission matrix of the multimode optical fiber can also vary over time due to factors such as fiber bending, fiber stress, or even temperature variations. Although the optical mesh configuration in this study was not performed in real-time due to speed limitations in the multichannel source measurement unit, progressive self-configuration with feedback has been previously demonstrated as a simple and rapid method to control the integrated optical mesh[37,38], which may be used to handle the arbitrary mode evolution and polarization rotation during optical fiber transmission. It is worth noting that the configuration speed of the photonic processor can be of the order of 10 μs when using thermal-optical phase shifters[48,49], and can be less than nanosecond when using electro-optical phase shifters[50]. This could potentially allow real-time management of the high-dimensional optical fiber systems using integrated photonic processors.

## Methods

### Multimode optical I/O design
The multimode grating coupler is designed for operation around 1550 nm wavelength range with a perfect vertical coupling configuration. 70-nm shallow etched holes are utilized as the low-index region for diffraction with a symmetrical pattern for the orthogonal polarizations. To reduce the coupling loss, chirped grating periods and hole diameter are optimized by genetic algorithm with effective medium

theory and 2D FDTD simulations. 3D FDTD simulations are performed to validate the coupling performances of all the high-order fiber modes. Four ADCs are used to (de)multiplex the $TE_0$ and $TE_1$ modes on chip. The relative phase shift is adjusted by a heater-based waveguide phase shifter with titanium-tungsten alloy (TiW) on the top. The two-mode FMF used in our experiment is fabricated by *OFS*.

### Photonic chip fabrication
The photonic chip is fabricated on a silicon-on-insulator (SOI) wafer with a 220 nm thick top silicon layer. The buried-oxide layer is 2 μm thick. Electro-beam lithography is used to define the device patterns, followed by dry reactive-ion etching process with a etch depth of 70 nm and a full etch. To protect the photonic circuits, a top cladding of silicon dioxide ($SiO_2$) with a thickness of 1.2 μm is used. Metallization is done using high-resistance titanium-tungsten alloy (TiW) for local heat generation and aluminum for electrical signal routing. A 300-nm thick $SiO_2$ passivation layer is used and selectively etched later over the aluminum pads for probing.

### MDM chip-to-chip data transmission
For high-speed inter-chip optical communications, the tunable laser (Santec TSL-570) is set at 1530 nm with an output power of 13 dBm injected into a LiNbO3 Mach-Zehnder modulator (AFR AM40). The optical modulator is driven by a BPG (Keysight 8045 A) with an RF amplifier (SHF S807C). The modulator optical signals are spatially decoupled by SMFs with a transmission distance of 2 km and 5 km and sent to the mode-selective fiber photonic lantern with a two-mode FMF. The integrated photonic processor is controlled by a multichannel source measurement unit (Nicslab XDAC-120U-R4G8) and

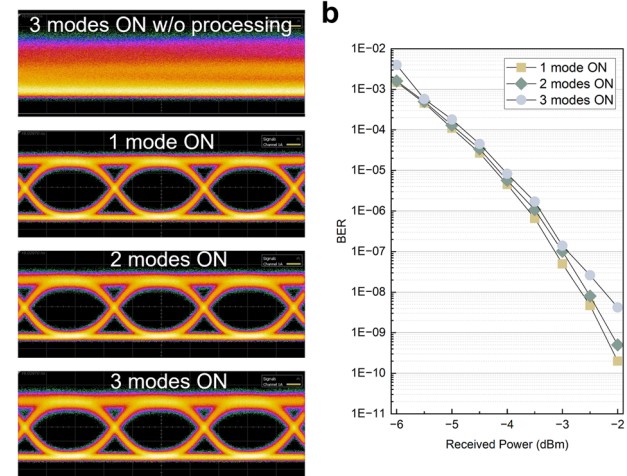

**Fig. 5 | Experimental results for MDM communications with three concurrent channels. a** Closed eye diagram when no optical processing is applied. A clean eye diagram can be retrieved but would be degraded slightly with additional concurrent mode activated. **b** BER with a power penalty when additional concurrent modes are switched on.

personal computer for the optical mesh configuration. An 8-channel optical power meter (Santec MPM-210H and MPM-215) is employed to read the optical powers from the fiber array. For eye diagram characterization at the receiver side, the optical signal is boosted by an erbium-doped fiber amplifier (EDFA, Amonics AEDFA-PA-35-B-FA) and sent to a 50-GHz PIN photodiode (Coherent XPDV2320R). The eye-diagram and bit error rate are obtained from a sampling oscilloscope (Keysight N1000A) and BERT (Keysight 8040A).

## Data availability

The data that support the findings of this study are included in the article and its supplementary information. Other data are available from the corresponding author upon request.

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

## Acknowledgements

Y.T. acknowledges the support from the National Natural Science Foundation of China (No. 62305277), the Guangzhou - HKUST(GZ) Joint Funding Program (No. 2023A03J0159), and the Start-up fund from the Hong Kong University of Science and Technology (Guangzhou). H.K.T. acknowledges the support from the Hong Kong Innovation and Technology Fund project (No. ITS/226/21FP). The authors acknowledge the Novel IC Exploration (NICE) Facility of HKUST(GZ) for technical support and Applied Nanotools Inc. for device fabrication.

## Author contributions

K.L., Z.C. and H.C. performed the experiment under the supervision of Y.T.; K.L. and Z.C. designed the photonic integrated processor; K.L. designed the printed circuit board with wire bonding; H.C. and K.L. developed the control algorithms and electronic control system; W.Z., Z.Z. and H.K.T. assisted in the device characterization; H.K.T. and Y.T. initiated the discussion. K.L., H.K.T. and Y.T. wrote the manuscript with contributions from all authors.

## Competing interests

The authors declare no competing interests.
