## [Peer Review File · Nature Communications]

Empowering high-dimensional optical fiber communications with integrated photonic processorsREVIEWER COMMENTS

Reviewer #1 (Remarks to the Author):

The authors present a novel approach to solving mode scrambling in multimode fiber. They demonstrate a chip-to-chip few-mode fiber system consisting of mode (de)multiplexers and a reconfigurable photonic processor, which successfully implements a mode descrambling process. The results of the study advance the field of research, however there are a few areas that should be addressed before the manuscript can be considered for publication.

The authors regrettably use “training” terminology throughout the manuscript to describe setting thermo-optic phase shifter values in the reconfigurable processor. This is highly misleading. The work is simply using metaheuristic algorithms, such as particle swarm optimization. The ‘training’ terminology incorrectly leads the reader to believe some form of machine learning is being used for the reconfiguration process.

There are a few grammatical and spelling errors scattered throughout the manuscript; please proofread the manuscript again to catch the remaining errors.

The authors state “However, such approaches originally developed for wireless communications require high-speed digital circuits with high-power consumption and large time latency when handling a data rate of over 100Gbaud, which severely hinders its potential use.” – How was this obtained, what is the reference for the threshold

The authors state that “similar coupling efficiencies for the y-polarized LP modes are guaranteed.” Where are the experimental measurements? To support these claims?

Mode descrambling training: How much time does it take to reconfigure the circuit and achieve selective decoupling? It is unclear that scaling these reconfiguration is practical in real systems.

Inter-chip multimode communication: What length PRBS is used for the bit error rate tests?

Discussion: authors fail to make the case for the broader impact of this work

S1: The authors mention using a genetic algorithm for optimization—more detail on this is needed to evaluate. In Figure S1, both tapered adiabatic directional couplers and symmetrical directional couplers are mentioned. But where are the tapered adiabatic directional couplers used?

S3: Simulation results point to minimal mode-dependent loss at a wavelength of 1560 nm. Why is a wavelength of 1530 nm used? Please walk through all the insertion losses; again case for scalability is not made.

S4: Is there any reason why the input grating coupler array is not compatible with an angled fiber array? Why are 3 modes launched into the photonic lantern and not all 6? Is this a limitation of the photonic lantern, or the experimental setup? Authors should clearly indicate the experimental limitations of the work and moderate the claims.

Reviewer #2 (Remarks to the Author):

The authors describe a spatial division multiplexing (SDM) communication system exploiting the transmission of data-streams carried by multiple modes in an optical fiber. Integrated optic structures are used to excite the modes in the fiber on the transmit side and to unravel them on the detection end of the optical link. Overall results are good with compelling agreement between

simulations and experiments on the behavior of the chip to fiber coupling elements. However, the results do not represent a breakthrough in materials or device physics but more an exciting piece of solid engineering and device optimization. Furthermore, though the link is operational and good results are obtained, the operational viability of the concept was not demonstrated. The total link loss is very high, demanding the application of an optical amplifier. Also, the optimization of the receiver chip is done in an offline mode. It is not clear how the system would handle a link distortion during operation. The cost and complexity seem too high for replacing conventional SDM systems with this new concept. The paper would profit from a comparison table with state-of-the-art solutions.

Some additional questions and comments:

- For readability, it would be helpful if the authors would already mention the presence of a grating coupler in the introduction to convert the planar 1D system of the integrated optic chip to the 2D field profile of the optical fiber.
- The authors describe SDM systems to cover multimode fiber communication. To my understanding an SDM system is related to communication channels multiplexed in space (multiple fibers or multicore fibers). Please check the definition and make sure modal multiplexing also falls in this category.
- The cost of the transceiver and the fiber is an important aspect in estimating the commercial viability of a communication concept. Here specialty fiber and transceiver elements are required. The cost-effective argument should be used with great care and to my opinion does not hold here.
- What is the temperature sensitivity of the optical chips?
- More details on the design of the grating coupler are required. How is the match between the fiber modes and the integrated optic waveguide field profile obtained.
- The training procedure was applied in an offline mode, continuous optimization during link operation is critical as mode coupling and interference in the fiber is changing over time.
- A reference should be added on 'particle-swarm optimization'

**Manuscript ID:** NCOMMS-23-50920

**Manuscript Title:** Empowering high-dimensional optical fiber communications with integrated
photonic processors

**Point-by-point response**

**Reviewer #1 (Comments to the Author):**

**Overall Remarks:**

**The authors present a novel approach to solving mode scrambling in multimode fiber. They**
**demonstrate a chip-to-chip few-mode fiber system consisting of mode (de)multiplexers and**
**a reconfigurable photonic processor, which successfully implements a mode descrambling**
**process. The results of the study advance the field of research, however there are a few areas**
**that should be addressed before the manuscript can be considered for publication.**

**Response:**

Thank you for your positive comments and valuable suggestions. We truly appreciate the
significant amount of time and effort you dedicated to this review. In the revised manuscript, we
have included additional design and experimental details and carefully proofread the manuscript.
Furthermore, the Discussion section has been rewritten to cover the advancement of our study,
experimental limitations, and future outlooks. Please find our point-by-point response attached.

**(1) Reviewer Comment:**

**The authors regrettably use “training” terminology throughout the manuscript to describe**
**setting thermo-optic phase shifter values in the reconfigurable processor. This is highly**
**misleading. The work is simply using metaheuristic algorithms, such as particle swarm**
**optimization. The ‘training’ terminology incorrectly leads the reader to believe some form**
**of machine learning is being used for the reconfiguration process.**

**(1) Response:**

Yes, we agree with the reviewer that the term “training” is misleading. The reconfigurable optical
mesh, consisting of tunable Mach-Zehnder interferometers, is appropriately configured using
simple maximization or minimization algorithms for mode descrambling process. We have
removed the term “training” and revised the manuscript, which are highlighted in red.

**(1) Revision:**

(Page 5, Line 110-114) “In order to mitigate the unknown inter-modal signal mixing, a
reconfigurable optical mesh with a dimensionality of eight can be programmed to apply the inverse
transformation of the fiber transmission matrix, thereby recovering the six orthogonal channels
permitted in the two-mode FMF.”

(Page 6, Line 140) “Mode descrambling experiment”

(Figure 3 caption) “c Evolution of the normalized transmission for eight output ports during
configuration of the optical mesh. d Bar chart of the initial random state, intermediate state, and
final state for 6 spatial and polarization channels during optical mesh configuration. e Bar chart of
another routing configuration of the optical mesh. The mode 1-6 refer to LP_{01-x}, LP_{01-y}, LP_{11a-x},
LP_{11a-y}, LP_{11b-x}, LP_{11b-y}.”

(Page 8, Line 157-159) “Figure 3c presents the evolution of normalized optical power from eight
output ports during configuration. It is evident that after around 120 iterations, selective decoupling
can be configured with a maximum crosstalk level suppression of >21.3 dB.”

(Page 9, Line 178-180) “After configuring integrated photonic processor, various orthogonal fiber
modes can be launched and routed to any desired output port to undo the signal mixing process.”

(Page 9, Line 194-185) “When two concurrent modes are launched without configuring the optical
mesh, the eye diagram becomes completely closed, as shown in Figure 4e.”

(Page 13, Line 277-279) “The integrated photonic processor is controlled by a multichannel source
measurement unit (Nicslab XDAC-120U-R4G8) and personal computer for the optical mesh
configuration.”

(Page 13, Line 297) “H.C. and K.L. developed the control algorithms and electronic control
system.”

**(2) Reviewer Comment:**

**There are a few grammatical and spelling errors scattered throughout the manuscript;**
**please proofread the manuscript again to catch the remaining errors.**

**(2) Response:**

Thank you for your valuable feedback and for bringing the grammatical and spelling errors to our
attention. We have carefully proofread the manuscript and try to improve the overall language and
editing of the manuscript.

**(2) Revision:**

Because so many changes were made to correct errors and improve the English, these revisions
are not highlighted here. Please refer to the red revised version for details.

**(3) Reviewer Comment:**

The authors state “However, such approaches originally developed for wireless
communications require high-speed digital circuits with high-power consumption and large
time latency when handling a data rate of over 100Gbaud, which severely hinders its
potential use.” – How was this obtained, what is the reference for the threshold

**(3) Response:**

Thank you for your valuable suggestions and comments. We recognize the importance of precision
and accuracy in scientific communication and apologize for the oversight in not properly
mentioning the threshold. We have removed the threshold, instead emphasizing the concerns on
power consumption and time latency at high data rates. We also mentioned that the increased
equalizer dimensionality and complexity when increasing the number of spatial modes.

**(3) Revision:**

(Page 2, Line 55-63) “Previous studies have demonstrated that this issue can be effectively
addressed through coherent communications and digital electronic multiple-input multiple-output
(MIMO) processing^{8,20}. However, this method, originally designed for wireless communications,
requires extremely high-speed digital circuits for optical fiber communications. The resulting high
power consumption and large time latency at high data rates become concerns. Moreover,
increasing the number of spatial channels would also rapidly increase the dimensionality of the
MIMO equalizer, thereby further increasing DSP complexity and impeding its potential utilization
21”

**(4) Reviewer Comment:**

The authors state that “similar coupling efficiencies for the y-polarized LP modes are
guaranteed.” Where are the experimental measurements? To support these claims?

**(4) Response:**

Thank you for your valuable suggestions and comments. In order to substantiate our claim and
enhance accuracy, we have included the experimental coupling efficiencies of the y-polarized LP
modes, as depicted Figure 2b. Furthermore, we utilized an infrared camera to capture the diffracted
field profiles from the multimode grating for all six LP modes, as shown in Figure 2c. We have
revised the manuscript as highlighted below.

**(4) Revision:**

(Page 6, Line 124-133) “Figure 2a shows the scanning electron microscope image of the 2D grating
coupler with 70-nm shallowly etched circular holes. The experimental coupling loss spectra of
LP₀₁, LP_{11a}, and LP_{11b} in the two orthogonal polarizations for a two-mode FMF are measured and
presented in Figure 2b. The x-polarized LP₀₁, LP_{11a}, and LP_{11b} exhibit a peak experimental

efficiency of -3.5 dB, -6.1 dB, and -4.3 dB at 1532 nm, 1517 nm, and 1515 nm respectively. As
 the 2D grating utilizes a symmetric structure, similar coupling efficiencies for the y-polarized
 modes can be obtained, measuring -3.9 dB at 1527 nm for LP_{01y} , -3.9 dB at 1517 nm for LP_{11a-y} ,
 and -6.1 dB at 1525 nm for LP_{11b-y} , respectively. To validate selective launching of the six LP
 modes through the multimode grating coupler, diffracted optical field of the grating are captured
 using a $10\times$ microscope objective and an infrared camera, as depicted in Figure 2c.”

 **Figure 2. Multimode grating coupler characterization and photonic processor design.** a Scanning electron
 microscope (SEM) image of the 2D grating coupler. b Experimental chip-to-fiber coupling efficiency spectra of the
 multimode grating coupler for various LP modes. c Optical field profile of the grating coupler captured by an infrared
 camera with a $10\times$ microscope objective when different fiber mode is selectively launched by the PIC_{TX}.

**(5) Reviewer Comment:**

**Mode descrambling training: How much time does it take to reconfigure the circuit and**
 **achieve selective decoupling? It is unclear that scaling these reconfiguration is practical in**
 **real systems.**

**(5) Response:**

Thank you for your valuable suggestions and comments. The real-time reconfiguration of the
 photonic processor is crucial for future applications, as mode mixing conditions can vary over time
 due to factors such as bending, deformation, and structural defects in a multimode optical fiber.

In our experiment, the configuration time may vary from tens of minutes to a few hours, depending
 on the speed of optimization convergence using our current setup. The slow configuration process
 is primarily due to the time required ($2\sim 3$ seconds each time) to reset all the output ports of our
 multichannel source measurement unit to control all the electrically tunable Mach-Zehnder
 interferometers. However, it is also worthwhile to mention that configuration speed can be
 significantly accelerated in the future, since the tuning speed of thermal optical phase shifters can
 be faster than $10 \mu\text{s}$, and less than sub-nanosecond by using electro-optic phase shifters. Real-time
 mode unmixing with the integrated photonic processor is potentially durable, which necessitates

high-speed control electronic circuits and feedback loops. The relevant insights are also included
in the Discussion section of the revised manuscript.

**(5) Revision:**

(Page 11, Line 242-252) “Real-time configuration of the integrated photonic processor is crucial
for high-dimensional fiber systems, as the transmission matrix of the multimode optical fiber can
also vary over time due to factors such as fiber bending, fiber stress, or even temperature variations.
Although the optical mesh configuration in this study was not performed in real time due to speed
limitations in the multichannel source measurement unit, progressive self-configuration with
feedback has been previously demonstrated as a simple and rapid method to control the integrated
optical mesh^{37,38}, which may be used to handle the arbitrary mode evolution and polarization
rotation during optical fiber transmission. It is worth noting that the configuration speed of the
photonic processor can be of the order of 10 μ s when using thermal-optical phase shifters^{48,49}, and
can be less than nanosecond when using electro-optical phase shifters⁵⁰. This could potentially
allow real-time management of the high-dimensional optical fiber systems using integrated
photonic processors.”

**(6) Reviewer Comment:**

**Inter-chip multimode communication: What length PRBS is used for the bit error rate tests?**

**(6) Response:**

Thank you for your valuable comments. The PRBS signal in the inter-chip communication
experiment has a length of $2^{15}-1$. We have included the PRBS signal length in the revised
manuscript, which is highlighted in red.

**(6) Revision:**

(Page 8, Line 170-173) “To evaluate the high-speed communication performance of the
reconfigurable photonic processor, 32Gbps non-return-to-zero (NRZ) signals are generated at the
transmitter side by a bit pattern generator (BPG) and an external LiNO₃ Mach-Zehnder modulator.
The pseudorandom binary sequence (PRBS) has a period length of $2^{15}-1$.”

**(7) Reviewer Comment:**

**Discussion: authors fail to make the case for the broader impact of this work**

**(7) Response:**

Thank you for your valuable suggestions and comments. We acknowledge the reviewer's
suggestion to enhance the Discussion section and emphasize its broader impact. We have revised
the Discussion section to incorporate a summary of the advancement of our study, experimental
limits, and future outlooks, as highlighted below.

**(7) Revision:**

(Page 10, Line 213-252) “We demonstrated a high-dimensional optical fiber communication
system enabled by a reconfigurable silicon photonic processor. This system incorporates selective
mode excitation and all-optical mode descrambling, achieved physically through the use of
multimode optical I/O and meshes of silicon MZIs. Without prior knowledge of the random mode
mixing and polarization rotation during the circular-core FMF transmission, we effectively
reconstruct six spatial and polarization channels, encompassing the full set of orthogonal channels
in a two-mode FMF for chip-to-chip optical fiber communications. As the mode launching and
unmixing are all conducted in the optical domain, our approach is expected to be generally capable
of managing communication systems utilizing various modulation formats. In real MDM optical
fiber system, mapping individual modes presents a challenge due to the unknown mode mixing
resulting from factors such as mode deformation, bending, or even structural imperfections in the
optical fiber. However, programmable photonic processors hold the potential to effectively
manage all the spatial channels prior to photodetection, thereby opening the door to numerous
high-dimensional optical fiber applications including communications, quantum networks,
imaging, and sensing.

Complete singular value decomposition (SVD) would be needed in the future to reduce the
crosstalk level and support more concurrent orthogonal data channels with a small BER penalty.
Although it would require a more complex optical mesh to execute non-unitary linear matrix
transformations, unraveling optical modes affected by significant mode-dependent loss and
differential mode group delay will be possible for multimode optical fiber systems⁴²⁻⁴⁴.

Increasing the number of optical channels can be achieved by harnessing the available optical
bandwidth of the integrated photonic processor. Each optical fiber mode can then be utilized in
conjunction with wavelength-division multiplexing to enhance the overall data throughput. Apart
from that, the number of involved spatial channels can also be increased by optimizing the
multimode optical I/O and the optical mesh. For instance, LP₂₁ mode can also be launched by
feeding two counterpropagating TE₁ modes with a relative phase shift of π , using the same
multimode grating coupler at the transmitter side in this work²⁶. To undo the signal mixing,
however, a non-unitary optical mesh would be required as not all of the degenerate modes in the
LP₂₁ group can be efficiently coupled back to the photonic chip. This may also require increasing
the dimension size of the integrated optical mesh or positioning it at the transmitter side^{43,47}.

Real-time configuration of the integrated photonic processor is crucial for high-dimensional fiber
systems, as the transmission matrix of the multimode optical fiber can also vary over time due to

factors such as fiber bending, fiber stress, or even temperature variations. Although the optical
mesh configuration in this study was not performed in real time due to speed limitations in the
multichannel source measurement unit, progressive self-configuration with feedback has been
previously demonstrated as a simple and rapid method to control the integrated optical mesh^{37,38},
which may be used to handle the arbitrary mode evolution and polarization rotation during optical
fiber transmission. It is worth noting that the configuration speed of the photonic processor can be
of the order of 10 μ s when using thermal-optical phase shifters^{48,49}, and can be less than
nanosecond when using electro-optical phase shifters⁵⁰. This could potentially allow real-time
management of the high-dimensional optical fiber systems using integrated photonic processors.”

**(8) Reviewer Comment:**

**S1: The authors mention using a genetic algorithm for optimization—more detail on this is**
**needed to evaluate. In Figure S1, both tapered adiabatic directional couplers and**
**symmetrical directional couplers are mentioned. But where are the tapered adiabatic**
**directional couplers used?**

**(8) Response:**

Thank you for your valuable suggestions and comments. We have included more details on the
grating coupler design method, grating parameters, and the genetic optimization process in the
revised manuscript. We sincerely apologize for the error in the figure S1 caption. The proposed
multimode optical I/O uses only tapered adiabatic directional couplers for (de)multiplexing of the
waveguide modes TE₀ and TE₁. To clearly illustrate the grating coupler design, we have
comprehensively revised the S1 section and re-plotted Figure S1, which is highlighted in red.

**(8) Revision:**

(Supplementary Note 1, Page 2-4) “In this work, a two-dimensional (2D) grating coupler is utilized
as the multimode optical I/O to facilitate the conversion between the planar waveguide mode on
the chip and the 2D field distribution of the optical fiber. Four linearly tapered mode size
converters and four tapered asymmetrical directional couplers (ADCs)^{1,2} are employed to convert
between all the planar waveguide modes and eight fundamental quasi-transverse-electric (TE)
modes prior to the optical mesh, as illustrated by Figure S1a.

Selective fiber mode launching at the transmitter can be realized via controlling the relative phase
difference between the two counterpropagating TE modes. Since the fundamental (TE₀) and first-
order (TE₁) modes exhibit similar effective indices in the silicon grating region with a width of 13
μ m, they can be efficiently diffracted out of the plane relying on the same grating structure. The
linearly polarized (LP) modes in a two-mode few-mode fiber (FMF) include LP_{01-x}, LP_{01-y},

LP11a-x, LP11a-y, LP11b-x, and LP11b-y. The corresponding launching conditions for each
optical fiber mode are summarized in Figure S1b.

The 2D grating coupler, with a width and length of 13.0 μm , is specially designed to match the
mode field diameter of 11.0 μm for the LP01 and LP11 modes in the two-mode graded-index few-
mode fiber (FMF) from OFS. The width of silicon waveguide is linearly tapered from 0.962 μm
to 13.0 μm by an adiabatic linear taper with a length of $L_{\text{taper}}=350 \mu\text{m}$ as shown in Figure S1a.
Figure S1c illustrates the schematic of the tapered ADC for (de)multiplexing the TE0-TE1 mode
on-chip. The integrated waveguide width w_1 , w_{2a} , and w_{2b} are 0.45 μm , 0.902 μm , and 0.962
239 μm , respectively. The waveguide gap g is designed as 0.2 μm and coupling length L is 33.6 μm .

The 2D grating coupler is formed by 70-nm shallowly etched circular holes, as depicted in Figure
S1d. The grating diffraction region is centrosymmetric to ensure uniform coupling performance for
all optical signals from the four orthogonally positioned waveguides. To achieve efficient and fully
vertical coupling, the grating periods are chirped and optimized using a genetic optimization
algorithm³ and finite-difference time domain (FDTD) simulations.

The optimization process evaluates coupling efficiency of the grating coupler using effective
medium theory^{4,5} and 2D FDTD simulation. This is advantageous as each simulation can be
completed in just a few seconds, made possible by two key factors. Firstly, the grating region is
symmetrical for the two polarizations, allowing for the consideration of only one polarization
during optimization. Additionally, mode TE0 and TE1 exhibit very similar effective indices in the
13.0 μm wide waveguide. As a result, the coupling efficiency of the fundamental TE0 mode in a
single polarization can serve as the figure of merit in the optimization iteration loop. The genetic
optimization algorithm is utilized to optimize both the grating period and circular hole diameter.
The optimal hole diameter is 343 nm. The chirped grating periods are depicted in Figure S1e.
Figure S1f illustrates the evolution diagram of coupling efficiency during the optimization process
using genetic algorithm.”

Figure S1 Design and optimization of the multimode optical I/O. **a** Schematic of multimode optical I/O, consisting of a two-dimensional (2D) waveguide grating, four linearly tapered mode size converters, and four tapered asymmetrical directional couplers (ADCs). **b** Illustration of fiber mode (de)multiplexing using the multimode grating coupler. **c** Schematic of the tapered ADC for TE_0 - TE_1 mode (de)multiplexing. **d** Top view and side view of the 2D grating coupler. Optimal grating periods and hole diameter obtained by genetic optimization. The symmetrical grating period is shown by the transparent windows. **e** Evolution diagram of normalized coupling efficiency during the optimization process.

**(9) Reviewer Comment:**

**S3: Simulation results point to minimal mode-dependent loss at a wavelength of 1560 nm.**
 **Why is a wavelength of 1530 nm used? Please walk through all the insertion losses; again**
 **case for scalability is not made.**

**(9) Response:**

Thank you for your valuable suggestions and comments. The experimental chip-to-fiber coupling
 efficiency spectra of the multimode grating coupler for various LP modes have a blue shift
 compared with the simulation results. In addition, the single mode grating coupler at the transmitter
 side and the receiver side have a coupling efficiency of about -8 dB at 1560 nm, as shown in Figure
 R1 below. The center wavelength is 1535 nm with a coupling loss of about -4.5 dB. Consequently,
 the optical system selected 1530 nm to conduct the chip-to-chip communication experiment to

reduce the total system loss while within the working wavelength range of the erbium-doped fiber
amplifier.

We have thoroughly summarized all the insertion loss in the revised Supplementary Information
Note 3. It is also important to mention that the overall insertion loss of the optical system can be
significantly reduced by optimizing the center wavelength of the signal mode grating coupler,
multimode grating coupler, and beam splitter/combiner in the Mach-Zehnder interferometers.
Furthermore, the Discussion section includes an outlook on the proposed system with loss
improvement and scalability to more data channels. We have added more discussions and
explanations to the revised manuscript, as highlighted in red.

**Figure R1.** Experimental coupling efficiency spectra of the single mode grating coupler.

**(9) Revision:**

(Page 8, Line 173-176) “Because the single mode grating coupler array in this work is also centered
at 1535 nm with a coupling loss of about -4.5 dB. The communication system is operated at 1530
290 nm to reduce the transmission loss and mode-dependent loss while within the working wavelength
range of the erbium-doped fiber amplifier.”

(Page 10, Supplementary Information Note 3) “In the system experiment, chip-to-chip high-speed
optical communications are conducted at a wavelength of 1530 nm. This choice of wavelength is
primarily motivated by three reasons: Firstly, the single-mode grating array exhibits a center
wavelength of 1535 nm with a coupling loss of -4.5 dB, while the coupling loss increases to -8
296 dB at 1560 nm. Secondly, the multimode grating coupler demonstrates low mode-dependent loss
at 1530 nm. Lastly, 1530 nm falls within the working wavelength range of the erbium-doped fiber
amplifier (EDFA) for optical amplification to compensate for optical power loss. In the future, the
center wavelength of all optical I/O devices and the optical mesh can be aligned to optimize the
overall loss of the communication system.”

At 1530 nm, the optimal insertion loss of the transmitter side is approximately 10 dB, with a loss
variation of roughly 3 dB for all spatial channels. An external tunable laser serves as the light
source. At the transmitter side, each single-mode input grating coupler experiences a transmission
loss of approximately 5 dB. The multimode grating coupler has a coupling loss of about 4~7 dB
for LP01 and LP11 mode groups, respectively. In the future, the insertion loss and loss variation
can be improved by optimizing the center wavelength of the input single-mode grating coupler
and multimode grating coupler.

At 1530 nm, the optimal total insertion loss of the receiver side is approximately 15 dB, with a
loss variation within 6 dB due to different routing paths with varying numbers of MZIs and the
mode-dependent loss of the multimode grating coupler. At the receiver side, the optical loss can
be broken down into 4~7 dB from the multimode grating coupler, 5 dB from the single mode
output grating coupler, around 0.35 dB for each multimode interferometer (MMI) (center
wavelength is designed at 1550 nm), about 4.5 dB from the waveguide and bends inside the optical
mesh. In the future, the total loss and loss variation can be mitigated by aligning the center working
wavelength of the MZIs to the multimode grating coupler.”

(Page 11, Line 227-241) “Complete singular value decomposition (SVD) would be needed in the
future to reduce the crosstalk level and support more concurrent orthogonal data channels with a
small BER penalty. Although it would require a more complex optical mesh to execute non-unitary
linear matrix transformations, unraveling optical modes affected by significant mode-dependent
loss and differential mode group delay will be possible for multimode optical fiber systems⁴²⁻⁴⁴.

Increasing the number of optical channels can be achieved by harnessing the available optical
bandwidth of the integrated photonic processor. Each optical fiber mode can then be utilized in
conjunction with wavelength-division multiplexing to enhance the overall data throughput. Apart
from that, the number of involved spatial channels can also be increased by optimizing the
multimode optical I/O and the optical mesh. For instance, LP21 mode can also be launched by
feeding two counterpropagating TE1 modes with a relative phase shift of π , using the same
multimode grating coupler at the transmitter side in this work²⁶. To undo the signal mixing,
however, a non-unitary optical mesh would be required as not all of the degenerate modes in the
LP21 group can be efficiently coupled back to the photonic chip. This may also require increasing
the dimension size of the integrated optical mesh or positioning it at the transmitter side^{43,47}.”

**(10) Reviewer Comment:**

**S4: Is there any reason why the input grating coupler array is not compatible with an angled**
**fiber array? Why are 3 modes launched into the photonic lantern and not all 6? Is this a**
**limitation of the photonic lantern, or the experimental setup? Authors should clearly indicate**
**the experimental limitations of the work and moderate the claims.**

**(10) Response:**

Thank you for your valuable suggestions and comments. Regrettably, the input grating coupler
array at the transmitter side is not spaced with an exact pitch size of 250 μm due to design errors,
as shown in Figure S4b. To facilitate simultaneous launching of the orthogonal fiber LP modes,
we utilize a fiber-based photonic lantern comprised of three single-mode fiber input ports.
Consequently, in our experiment, three modes can be launched using the three physical single-
mode fiber input ports. In addition, we think it is also meaningful to include the experimental
results of launching two concurrent data channels. The integrated optical mesh will be configured
to retrieve the optical signals from two orthogonal fiber modes, including the combination of LP₀₁
with LP_{11a}, LP₀₁ with LP_{11b}, and LP_{11a} with LP_{11b}. Additional experimental results have been
included in Figure 4. Due to lack of optical switch or laser source, the optical mesh cannot be
simultaneously configured for three concurrent channels offline. Thus, we used additional single-
mode fibers to obtain three spatially decoupled data channels for the communication experiment,
as illustrated in Figure S4d. Increasing the number of spatial channels would also require a better
inter-modal crosstalk to avoid severe BER penalty, which necessitates more complex optical mesh
to execute the non-unitary transformations in the future.

We agree with the reviewer's suggestion that experimental limitations should be clearly indicated,
and claims should be moderated in the manuscript. We have incorporated these changes into the
revised manuscript, as highlighted in red.

**(10) Revision:**

(Page 1, Line 21-27) “Here we show that a high-dimensional optical fiber communication system
can be implemented by a reconfigurable integrated photonic processor, featuring kernels of
multichannel mode multiplexing transmitter and all-optical descrambling receiver. Effective mode
management can be achieved through the configuration of the integrated optical mesh. Inter-chip
MDM optical communications involving six spatial- and polarization modes was realized, despite
the presence of unknown mode mixing and polarization rotation in the circular-core fiber.”

(Page 10, Line 213-252) “We demonstrated a high-dimensional optical fiber communication
system enabled by a reconfigurable silicon photonic processor. This system incorporates selective
mode excitation and all-optical mode descrambling, achieved physically through the use of
multimode optical I/O and meshes of silicon MZIs. Without prior knowledge of the random mode
mixing and polarization rotation during the circular-core FMF transmission, we effectively
reconstruct six spatial and polarization channels, encompassing the full set of orthogonal channels
in a two-mode FMF for chip-to-chip optical fiber communications. As the mode launching and
unmixing are all conducted in the optical domain, our approach is expected to be generally capable
of managing communication systems utilizing various modulation formats. In real MDM optical
fiber system, mapping individual modes presents a challenge due to the unknown mode mixing
resulting from factors such as mode deformation, bending, or even structural imperfections in the

optical fiber. However, programmable photonic processors hold the potential to effectively
manage all the spatial channels prior to photodetection, thereby opening the door to numerous
high-dimensional optical fiber applications including communications, quantum networks,
imaging, and sensing.

Complete singular value decomposition (SVD) would be needed in the future to reduce the
crosstalk level and support more concurrent orthogonal data channels with a small BER penalty.
Although it would require a more complex optical mesh to execute non-unitary linear matrix
transformations, unraveling optical modes affected by significant mode-dependent loss and
differential mode group delay will be possible for multimode optical fiber systems⁴²⁻⁴⁴.

Point-by-point response

Reviewer #2 (Comments to the Author):

Overall Remarks:

The authors describe a spatial division multiplexing (SDM) communication system exploiting the transmission of data-streams carried by multiple modes in an optical fiber. Integrated optic structures are used to excite the modes in the fiber on the transmit side and to unravel them on the detection end of the optical link. Overall results are good with compelling agreement between simulations and experiments on the behavior of the chip to fiber coupling elements. However, the results do not represent a breakthrough in materials or device physics but more an exciting piece of solid engineering and device optimization. Furthermore, though the link is operational and good results are obtained, the operational viability of the concept was not demonstrated. The total link loss is very high, demanding the application of an optical amplifier. Also, the optimization of the receiver chip is done in an offline mode. It is not clear how the system would handle a link distortion during operation. The cost and complexity seem too high for replacing conventional SDM systems with this new concept. The paper would profit from a comparison table with state-of-the-art solutions.

Response:

Thank you for your insightful comments and we greatly appreciate the substantial amount of time and effort that you dedicated to this review. We agree with the reviewer that the main concerns should be addressed including the link loss, real-time handling of modal crosstalk, and the overall system cost and complexity. State-of-the-art comparisons should also be included.

(1) Clarification of novelty: In this study, we propose that addressing the unknown mode scrambling in a multimode optical fiber system can be effectively achieved through integrated photonic processors. The 2D fiber modes can be converted into planar waveguide modes for on-chip optical signal processing before photodetection. To the best of our knowledge, for the first time, the complete set of six orthogonal channels in a two-mode FMF can be selectively launched and retrieved without using bulk optics approach. As the mode launching and unmixing are all performed in the optical domain, our approach is generally capable of managing communication systems utilizing various modulation formats. Additionally, the proposed photonic processor is based on the silicon photonics platform, enabling mass production through the mature CMOS fabrication process. We have revised the introduction and discussion of our manuscript, as highlighted in red.

(2) Concern regarding link loss: In our experiment, the optical signal goes through two single-mode grating couplers and two multimode grating couplers due to non-integrated light sources and photodiodes are used. The PIN photodiode used at the receiver side does not have a trans-impedance amplifier. Consequently, an optical amplifier is needed in our system-level

demonstration. It is worth noting that the overall insertion loss can be significantly reduced in the
future. A detailed breakdown of optical losses has been included in the revised Supplementary
Information Note 3. The reduction of total insertion loss can be achieved through the alignment of
the center wavelength of the signal mode grating coupler, multimode grating coupler, and beam
splitter/combiner in the Mach-Zehnder interferometers. In addition, we also included a comparison
table for the multimode grating coupler with prior art in Table S1. Six orthogonal fiber modes can
be supported efficiently in our work. We have revised the manuscript with revisions highlighted
in red.

**(3) Concern regarding offline configuration of the photonic processor:** in our experiment, the
slow configuration process is primarily due to the time required (2~3 seconds each time) to reset
all the output ports of our multichannel source measurement unit to control the electrically tunable
Mach-Zehnder interferometers. However, it is important to note that configuration speed can be
significantly accelerated, since the tuning speed of thermal optical phase shifters can be faster than
faster than 10 μ s, and less than sub-nanosecond by using electro-optic phase shifters. Real-time
self-configuration of the integrated photonic processor may be durable, which necessitates control
electronic circuits and feedback loops. We have revised the Discussion to discuss our experimental
limitations and future outlooks, as highlighted in red.

**(4) Concern regarding the cost and complexity:** the integrated photonic processor proposed in
this work can directly work with few-mode fibers without using any bulk optics to launch and
retrieve the six orthogonal fiber modes. Compared to the prior works using free space optics, we
believe our work can reduce the system complexity and cost, leveraging the integrated photonics
platform with mass production ability. In the meantime, the proposed photonic processor can
address the strong mode crosstalk in a circular-core multimode fiber in the optical domain, which
is expected to work with different modulation formats. The minimal crosstalk may obviate the
need for digital signal processing or necessitate a sparse MIMO equalizer, thereby reducing the
complexity of the MIMO equalizer in the digital signal processing domain. We agree with the
reviewer that including a comparison with state-of-the-art solutions would enhance our manuscript.
A comprehensive and fair system-level comparison is challenging as the power consumption and
complexity of using digital signal processing also vary based on modulation formats and data rate.
Nevertheless, we tried to include a comparison table S1 for the multimode optical I/O used in our
work to clarify the advancement of our study. The relevant insights are also included in the
Discussion section of the revised manuscript.

**Revision:**

(Page 10, Line 213-252) “We demonstrated a high-dimensional optical fiber communication
system enabled by a reconfigurable silicon photonic processor. This system incorporates selective
mode excitation and all-optical mode descrambling, achieved physically through the use of
multimode optical I/O and meshes of silicon MZIs. Without prior knowledge of the random mode
mixing and polarization rotation during the circular-core FMF transmission, we effectively
reconstruct six spatial and polarization channels, encompassing the full set of orthogonal channels

in a two-mode FMF for chip-to-chip optical fiber communications. As the mode launching and
unmixing are all conducted in the optical domain, our approach is expected to be generally capable
of managing communication systems utilizing various modulation formats. In real MDM optical
fiber system, mapping individual modes presents a challenge due to the unknown mode mixing
resulting from factors such as mode deformation, bending, or even structural imperfections in the
optical fiber. However, programmable photonic processors hold the potential to effectively
manage all the spatial channels prior to photodetection, thereby opening the door to numerous
high-dimensional optical fiber applications including communications, quantum networks,
imaging, and sensing.

Complete singular value decomposition (SVD) would be needed in the future to reduce the
Crosstalk level and support more concurrent orthogonal data channels with a small BER penalty.
Although it would require a more complex optical mesh to execute non-unitary linear matrix
transformations, unraveling optical modes affected by significant mode-dependent loss and
differential mode group delay will be possible for multimode optical fiber systems⁴²⁻⁴⁴.

Increasing the number of optical channels can be achieved by harnessing the available optical
bandwidth of the integrated photonic processor. Each optical fiber mode can then be utilized in
conjunction with wavelength-division multiplexing to enhance the overall data throughput. Apart
from that, the number of involved spatial channels can also be increased by optimizing the
multimode optical I/O and the optical mesh. For instance, LP₂₁ mode can also be launched by
feeding two counterpropagating TE₁ modes with a relative phase shift of π , using the same
multimode grating coupler at the transmitter side in this work²⁶. To undo the signal mixing,
however, a non-unitary optical mesh would be required as not all of the degenerate modes in the
LP₂₁ group can be efficiently coupled back to the photonic chip. This may also require increasing
the dimension size of the integrated optical mesh or positioning it at the transmitter side^{43,47}.

Real-time configuration of the integrated photonic processor is crucial for high-dimensional fiber
systems, as the transmission matrix of the multimode optical fiber can also vary over time due to
factors such as fiber bending, fiber stress, or even temperature variations. Although the optical
mesh configuration in this study was not performed in real time due to speed limitations in the
multichannel source measurement unit, progressive self-configuration with feedback has been
previously demonstrated as a simple and rapid method to control the integrated optical mesh^{37,38},
which may be used to handle the arbitrary mode evolution and polarization rotation during optical
fiber transmission. It is worth noting that the configuration speed of the photonic processor can be
of the order of 10 μ s when using thermal-optical phase shifters^{48,49}, and can be less than
nanosecond when using electro-optical phase shifters⁵⁰. This could potentially allow real-time
management of the high-dimensional optical fiber systems using integrated photonic processors.”

**Table S1.** Comparison of integrated multimode optical I/O for multi-mode fibers

Design	Fiber type	Num. of spatial channels	Experimental coupling efficiency [dB]
Grating array ⁶	2-mode circular core FMF	6	LP ₀₁ -LP ₁₁ : <-20
Grating array ⁷	2-mode circular core FMF	6	LP ₀₁ -LP ₁₁ : -23
2D grating ⁸	2-mode circular core FMF	8	LP ₀₁ : -22
2D grating ⁹	2-mode circular core FMF	4	LP ₀₁ : -4.9, LP ₁₁ : -6.1
Grating coupler ¹⁰	2-mode circular core FMF	2	LP ₀₁ : -1.36, LP ₁₁ : -2.21
Grating array ¹¹	2-mode circular core FMF	6	LP ₀₁ : -5.2, LP ₁₁ : -9.0
Edge coupler ¹²	2-mode circular core FMF	2	LP ₀₁ : -13.2, LP ₁₁ : -12.5
2D grating This work	2-mode circular core FMF	6	LP ₀₁ : -3.5, LP ₁₁ : -6.1

**(1) Reviewer Comment:**

**For readability, it would be helpful if the authors would already mention the presence of a**
**grating coupler in the introduction to convert the planar 1D system of the integrated optic**
**chip to the 2D field profile of the optical fiber.**

**(1) Response:**

Thank you for your valuable suggestions. We agree with the reviewer that field conversion should
be mentioned in the introduction to improve readability. We have added the suggested information
to the introduction in the revised manuscript, as highlighted in red.

**(1) Revision:**

(Page 3, Line 64-68) **“By transforming the 2D field profile of the optical fiber into the planar**
**waveguide modes on the integrated photonic chip, such as through grating couplers^{17,22-26},**
**photonic processors present a promising alternative technology for managing the high-dimensional**
**optical fiber system, especially on the silicon photonics platform, which offers low-cost, high-**
**volume manufacturing with CMOS compatibility²⁷⁻²⁹.”**

**(2) Reviewer Comment:**

**The authors describe SDM systems to cover multimode fiber communication. To my**
**understanding an SDM system is related to communication channels multiplexed in space**
**(multiple fibers or multicore fibers). Please check the definition and make sure modal**
**multiplexing also falls in this category.**

**(2) Response:**

Thank you for your valuable suggestions. We agree with the reviewer that it is necessary to revisit
whether the modal multiplexing also falls in the space-division multiplexing systems. We have
checked the relevant publications as well as the recent review about the SDM systems. In Refs [1,
5, 6], SDM fibers include the multicore fibers, multimode/few-mode fibers, and few-mode
multicore fibers. Therefore, we have revised the manuscript and included the references for the
fiber and technology category definitions, which is highlighted in red below.

**(2) Revision:**

(Page 2, Line 30-35) “The spatial dimension of optical fibers is an unexploited resource for
enhancing its information-transmission capacity¹⁻⁴. Space-division multiplexing (SDM), whereby
multiple data signals are multiplexed into the different spatial channels, has attracted much
research interest including the use of multiple single-mode cores sharing a common cladding or
multiple modes with different mode field patterns in a SDM optical fiber. SDM fiber can thus be
classified into multi-core fiber (MCF), few-mode fiber (FMF), and multi-mode fiber (MMF)^{1,5,6}”

[1] Puttnam, B. J., Rademacher, G. & Luís, R. S. Space-division multiplexing for optical fiber
communications. Optica, OPTICA 8, 1186–1203 (2021).

[5] van Uden, R. G. H. et al. Ultra-high-density spatial division multiplexing with a few-mode
multicore fibre. Nature Photon 8, 865–870 (2014).

[6] Fontaine, N. K. et al. Photonic Lanterns, 3-D Waveguides, Multiplane Light Conversion, and
Other Components That Enable Space-Division Multiplexing. Proceedings of the IEEE 110, 1821–
1834 (2022).

**(3) Reviewer Comment:**

**The cost of the transceiver and the fiber is an important aspect in estimating the commercial**
**viability of a communication concept. Here specialty fiber and transceiver elements are**
**required. The cost-effective argument should be used with great care and to my opinion does**
**not hold here.**

**(3) Response:**

Thank you for your valuable suggestions. We agree with the reviewer that we should be very
careful about the claim of being "cost-effective." Although the demonstrated integrated photonic
processor approach potentially offers a low-cost solution for high-dimensional optical fiber
systems compared with bulky optical components, primarily benefiting from volume
manufacturing with CMOS compatibility, the relatively low cost and complexity do not
necessarily lead to a "cost-effective" claim. Therefore, we have revised the manuscript and now

only claim the potential for lower cost rather than cost-effectiveness, which is highlighted in red
below.

**(3) Revision:**

(Page 2, Line 37-41) “Two major challenges are associated with MDM optical fiber systems,
including the lack of low-cost and scalable mode (de)multiplexers that can generate or decouple
multiple orthogonal fiber modes, and the substantial energy consumption and large time latency
incurred in descrambling high-speed optical signals using electronic digital signal processing⁷”

(Page 2, Line 43-46) “Although substantial advancements have been achieved in fiber-based
photonic lanterns or laser-inscribed waveguides^{5,6,15}, a compact and low-cost approach is desired,
especially for short-reach optical communications within data centers where cost and footprint are
crucial factors.”

**(4) Reviewer Comment:**

**What is the temperature sensitivity of the optical chips?**

**(4) Response:**

Thank you for your valuable comments. In our experiment, we utilized thermal-optical phase
shifters on both the transmitter and receiver sides to facilitate mode generation and the signal
unscrambling process, necessitating the use of a thermoelectric cooler (TEC) for thermal
stabilization of the integrated photonic processor. Figure R2 depicts the crosstalk levels against
the temperature variation of the optical mesh at the receiver side after configuration. The crosstalk
values were measured three times at each temperature, and the upper and lower crosstalk values
are indicated with error bars. We can see that crosstalk degradation can be less than 3dB when the
temperature variation is within 2.5°C. Most importantly, the optical mesh can be reconfigured at
different temperatures to achieve comparable crosstalk levels in our experiment. Therefore,
feedback loops and high-speed control electronic circuits may be utilized in the future to handle
temperature variation of the photonic chip. We have revised the manuscript and included the
temperature sensitivity of the crosstalk, as well as the discussion of future insights.

**Figure R2.** Temperature sensitivity of crosstalk at the receiver side after configuration of the optical mesh.

**(4) Revision:**

(Page 7, Line 148-149) “A thermoelectric cooler (TEC) is used for thermal stabilization of the
 integrated photonic processor at the receiver side.”

(Page 8, Line 165-168) “Temperature sensitivity of the integrated photonic processor is also
 evaluated. When the temperature variation is kept below 2.5°C, the increase in inter-modal
 crosstalk can be limited to 3 dB. The optical mesh can be reconfigured to avoid crosstalk
 degradation, highlighting the need for real-time configuration in the future.”

(Page 12, Line 242-252) “Real-time configuration of the integrated photonic processor is crucial
 for high-dimensional fiber systems, as the transmission matrix of the multimode optical fiber can
 also vary over time due to factors such as fiber bending, fiber stress, or even temperature variations.
 Although the optical mesh configuration in this study was not performed in real time due to speed
 limitations in the multichannel source measurement unit, progressive self-configuration with
 feedback has been previously demonstrated as a simple and rapid method to control the integrated
 optical mesh^{37,38}, which may be used to handle the arbitrary mode evolution and polarization
 rotation during optical fiber transmission. It is worth noting that the configuration speed of the
 photonic processor can be of the order of 10 μs when using thermal-optical phase shifters^{48,49}, and
 can be less than nanosecond when using electro-optical phase shifters⁵⁰. This could potentially
 allow real-time management of the high-dimensional optical fiber systems using integrated
 photonic processors.”

**(5) Reviewer Comment:**

**More details on the design of the grating coupler are required. How is the match between**
**the fiber modes and the integrated optic waveguide field profile obtained.**

**(5) Response:**

Thank you for your valuable suggestions. We have included more details on the grating coupler
design method, grating parameters, the genetic optimization process to improve the coupling
efficiency. To clearly illustrate the grating coupler design, we have revised Figure 2 and
Supplementary Note 1 section, which is highlighted in red.

**(5) Revision:**

(Page 6, Line 124-133) **“Figure 2a shows the scanning electron microscope image of the 2D grating**
**coupler with 70-nm shallowly etched circular holes. The experimental coupling loss spectra of**
**LP₀₁, LP_{11a}, and LP_{11b} in the two orthogonal polarizations for a two-mode FMF are measured and**
**presented in Figure 2b. The x-polarized LP₀₁, LP_{11a}, and LP_{11b} exhibit a peak experimental**
**efficiency of -3.5 dB, -6.1 dB, and -4.3 dB at 1532 nm, 1517 nm, and 1515 nm respectively. As**
**the 2D grating utilizes a symmetric structure, similar coupling efficiencies for the y-polarized**
**modes can be obtained, measuring -3.9 dB at 1527 nm for LP_{01y}, -3.9 dB at 1517 nm for LP_{11a-y},**
**and -4.3 dB at 1525 nm for LP_{11b-y}, respectively. To validate selective launching of the six LP**
**modes through the use of the multimode grating coupler, diffracted optical field of the grating are**
**captured using a 10× microscope objective and an infrared camera, as depicted in Figure 2c.”**

**Figure 2. Multimode grating coupler characterization and photonic processor design. a** Scanning electron
**microscope (SEM) images of the 2D grating coupler. b** Experimental chip-to-fiber coupling efficiency spectra of the
**multimode grating coupler for various LP modes. c** Optical field profile of the grating coupler captured by an infrared
**camera with a 10× microscope objective when different fiber mode is selectively launched by the PIC_{TX}.**

(Supplementary Note 1, Page 2-4) **“In this work, a two-dimensional (2D) grating coupler is utilized**
**as the multimode optical I/O to facilitate the conversion between the planar waveguide mode on**
**the chip and the 2D field distribution of the optical fiber. Four linearly tapered mode size**

converters and four tapered asymmetrical directional couplers (ADCs)^{1,2} are employed to convert
between all the planar waveguide modes and eight fundamental quasi-transverse-electric (TE)
modes prior to the optical mesh, as illustrated by Figure S1a.

Selective fiber mode launching at the transmitter can be realized via controlling the relative phase
difference between the two counterpropagating TE modes. Since the fundamental (TE₀) and first-
order (TE₁) modes exhibit similar effective indices in the silicon grating region with a width of 13
637 μm, they can be efficiently diffracted out of the plane relying on the same grating structure. The
638 linearly polarized (LP) modes in a two-mode few-mode fiber (FMF) include LP_{01-x}, LP_{01-y}, LP_{11a-}
x, LP_{11a-y}, LP_{11b-x}, and LP_{11b-y}. The corresponding launching conditions for each optical fiber mode
are summarized in Figure S1b.

The 2D grating coupler, with a width and length of 13.0 μm, is specially designed to match the
mode field diameter of 11.0 μm for the LP₀₁ and LP₁₁ modes in the two-mode graded-index few-
mode fiber (FMF) from OFS. The width of silicon waveguide is linearly tapered from 0.962 μm
to 13.0 μm by an adiabatic linear taper with a length of L_{taper}=350 μm as shown in Figure S1a.
Figure S1c illustrates the schematic of the tapered ADC for (de)multiplexing the TE₀-TE₁ mode
on-chip. The integrated waveguide width w₁, w_{2a}, and w_{2b} are 0.45 μm, 0.902 μm, and 0.962
647 μm, respectively. The waveguide gap g is designed as 0.2 μm and coupling length L is 33.6 μm.

The 2D grating coupler is formed by 70-nm shallowly etched circular holes, as depicted in Figure
S1d. The grating diffraction region is centrosymmetric to ensure uniform coupling performance for
all optical signals from the four orthogonally positioned waveguides. To achieve efficient and fully
vertical coupling, the grating periods are chirped and optimized using a genetic optimization
algorithm³ and finite-difference time domain (FDTD) simulations.

The optimization process evaluates coupling efficiency of the grating coupler using effective
medium theory^{4,5} and 2D FDTD simulation. This is advantageous as each simulation can be
completed in just a few seconds, made possible by two key factors. Firstly, the grating region is
symmetrical for the two polarizations, allowing for the consideration of only one polarization
during optimization. Additionally, mode TE₀ and TE₁ exhibit very similar effective indices in the
13.0 μm wide waveguide. As a result, the coupling efficiency of the fundamental TE₀ mode in a
single polarization can serve as the figure of merit in the optimization iteration loop. The genetic
optimization algorithm is utilized to optimize both the grating period and circular hole diameter.
The optimal hole diameter is 343 nm. The chirped grating periods are depicted in Figure S1e.
Figure S1f illustrates the evolution diagram of coupling efficiency during the optimization process
using genetic algorithm.”

**Figure S1 Design and optimization of the multimode optical I/O. a** Schematic of multimode optical I/O, consisting
 of a two-dimensional (2D) waveguide grating, four linearly tapered mode size converters, and four tapered
 asymmetrical directional couplers (ADCs). **b** Illustration of fiber mode (de)multiplexing using the multimode grating
 coupler. **c** Schematic of the tapered ADC for TE₀-TE₁ mode (de)multiplexing. **e** Top view and side view of the 2D
 grating coupler. Optimal grating periods and hole diameter obtained by genetic optimization. The symmetrical grating
 period is shown by the transparent windows. **f** Evolution diagram of normalized coupling efficiency during the
 optimization process.

**(6) Reviewer Comment:**

The training procedure was applied in an offline mode, continuous optimization during link
 operation is critical as mode coupling and interference in the fiber is changing over time.

**(6) Response:**

Thank you for your valuable suggestions. The real-time reconfiguration of the photonic processor
 is crucial for applications, as mode mixing conditions can vary over time due to factors such as

bending, deformation, and structural defects in a multimode optical fiber. In our experiment,
offline configuration process is primarily due to the time required (2~3 seconds each time) to reset
all the output ports of our multichannel source measurement unit to control the electrically tunable
Mach-Zehnder interferometers. However, it is important to note that configuration speed can be
significantly accelerated, since the tuning speed of thermal optical phase shifters can be faster than
faster than 10 μ s, and less than sub-nanosecond by using electro-optic phase shifters. Real-time
mode unmixing with the integrated photonic processor is durable, which necessitates high-speed
control electronic circuits and feedback loops. The relevant insights are also included in the
Discussion section of the revised manuscript.

**(6) Revision:**

(Page 12, Line 242-252) “Real-time configuration of the integrated photonic processor is crucial
for high-dimensional fiber systems, as the transmission matrix of the multimode optical fiber can
also vary over time due to factors such as fiber bending, fiber stress, or even temperature variations.
Although the optical mesh configuration in this study was not performed in real time due to speed
limitations in the multichannel source measurement unit, progressive self-configuration with
feedback has been previously demonstrated as a simple and rapid method to control the integrated
optical mesh^{37,38}, which may be used to handle the arbitrary mode evolution and polarization
rotation during optical fiber transmission. It is worth noting that the configuration speed of the
photonic processor can be of the order of 10 μ s when using thermal-optical phase shifters^{48,49}, and
can be less than nanosecond when using electro-optical phase shifters⁵⁰. This could potentially
allow real-time management of the high-dimensional optical fiber systems using integrated
photonic processors.”

**(7) Reviewer Comment:**

**A reference should be added on ‘particle-swarm optimization’**

**(7) Response:**

Thank you for your valuable suggestions. We agree with the reviewer that a reference should be
included on the “particle-swarm optimization”. We have made the necessary revisions to the
manuscript, and these changes are highlighted in red below.

**(7) Revision:**

(Page 7, Line 150-152) “In the experiments, we have employed multichannel source measurement
units (SMUs) and the particle-swarm optimization⁴³ algorithm to optimize the drive voltage of
phase shifters within the optical mesh $U(8)$.”

43. Wang, D., Tan, D. & Liu, L. Particle swarm optimization algorithm: an overview. Soft Comput
22, 387–408 (2018).

REVIEWERS' COMMENTS

Reviewer #1 (Remarks to the Author):

the authors addressed the review

Reviewer #2 (Remarks to the Author):

Thank you for carefully responding to the comments and suggestions made.